# Increases in the Macrolide Resistance of *Mycoplasma genitalium* and the Emergence of the A2058T Mutation in the 23S rRNA Gene: Clonal Spread?

**DOI:** 10.3390/antibiotics11111492

**Published:** 2022-10-27

**Authors:** Luis Piñeiro, Pedro Idigoras, Maitane Arrastia, Ayla Manzanal, Iñigo Ansa, Gustavo Cilla

**Affiliations:** 1Microbiology Department, Donostia University Hospital, 20014 Donostia, Spain; 2Biodonostia Health Research Institute, 20014 Donostia, Spain

**Keywords:** *Mycoplasma genitalium*, antibiotic resistance surveillance, A2058T mutation emergence, genetic characterisation

## Abstract

The management of *Mycoplasma genitalium* sexually transmitted infection (STI) is hindered by increasing resistance to the recommended antibiotics, macrolides and quinolones, worldwide. In Gipuzkoa (Basque Country, Spain), macrolide and quinolone resistance rates in 2014–2018 were reported as <20% and <10%, respectively. The aims of this study were to compare these rates with those in 2019–2021 and analyse the genetic and epidemiological features of the strains and cases associated with striking changes in the resistance trends. Resistance to macrolides (*n* = 1019) and quinolones (*n* = 958) was studied, analysing mutations in 23S rRNA and *parC*/*gyrA* genes, respectively. The rate of macrolide resistance increased from 17.3% in 2014–2018 to 32.1% in 2019–2021, as much in the more prevalent A2058/2059G mutations (16.6–27.8%) as in the emergent A2058T mutations (0.5–4.1%) but with differences in the odds ratios and the relative risk increase between A2058T and A2058/2059G mutations. MG191 adhesin and MG309 lipoprotein of the 27 emergent strains detected with A2058T mutations were amplified, sequenced, and typed using phylogenetic and variable number tandem repeat analysis, respectively. Genetic clonal spread was ruled out, but most of the A2058T cases were men who had sex with men (24/27) with a history of STI and antibiotic treatments (19/27). No changes were observed in quinolone resistance trends, but the rate of resistance to both antibiotics rose from 2.9% to 8.3%, especially in cases with A2058T mutations. The genetic characterisation of strains and epidemiological surveillance of cases are needed to detect populations at increased risk of treatment failure in this infection.

## 1. Introduction

Infection by *Mycoplasma genitalium* can cause urethritis in men and cervicitis and pelvic inflammatory disease in women and is a growing public health problem worldwide [1]. It has usually been related to persistent and/or recurrent sexually transmitted infections (STIs), but in many cases, such apparent relationships might be attributable to late diagnosis and/or incorrect treatment. Given that it is a small bacterial species without a cell wall, its isolation requires special culture media and can be difficult and slow, and hence, this infection should be diagnosed using nucleic acid amplification techniques (NAATs).

In recent years, the diagnostic and therapeutic strategies for *M. genitalium* infection proposed in the main STI guidelines have been changing and adapting to increases in the rates of macrolide resistance. These rates often exceed 40%, especially in populations at the highest risk of STIs and previously treated with azithromycin [2,3,4,5,6]. In line with this, the International Union against Sexually Transmitted Infections (IUSTI) continues recommending, in Europe, the use of oral azithromycin (500 mg) for the first day of treatment followed by 250 mg/d for 4 days in the case of macrolide-susceptible *M. genitalium* infections and oral moxifloxacin (400 mg) for 7 or 14 days as an alternative in macrolide-resistant or complicated infections, respectively [1]. In contrast, the Australian STI Management Guidelines and the US Centers for Disease Control and Prevention have started to recommend sequential targeted therapy with oral doxycycline 100 mg q 12 h for 7 days, followed by azithromycin for susceptible strains and sitafloxacin [7] or moxifloxacin [8] for resistant strains. Given this, for the adequate management of *M. genitalium* infection, which is associated with symptoms that are similar to other STIs but which should be treated with different antibiotics (for example, *Chlamydia trachomatis* with doxycycline or a single 1-g dose of azithromycin and *Neisseria gonorrhoeae* with 500 mg of ceftriaxone), this microorganism should be included in the differential diagnosis of symptomatic STIs from the outset, using syndromic NAATs. Further, to start targeted therapy, macrolide susceptibility should be investigated early using NAATs that detect mutations in the 23S rRNA gene associated with macrolide resistance (mainly A2058G and A2059G and, to a lesser extent, A2058T, A2059C and A2058C). Although doxycycline therapy has limited cure rates (~30%), sequential targeted therapy could be a useful strategy in contexts with high macrolide resistance rates and if a macrolide susceptibility test could not be performed early, allowing the initial use of doxycycline to lower the *M. genitalium* load [5]. The subsequent analysis of mutations in the *parC* and *gyrA* genes encoding the quinolone-resistance determining region (QRDR) enables treatment to be adjusted as appropriate, although its concordance with phenotypic and in vivo resistance is variable [9,10,11]. G248T mutations in the *parC* gene (S83I aminoacid change) have been associated with higher moxifloxacin treatment failures, while the absence of mutations at *parC* S83/D87 was highly predictive of cure (>98%).

Surveillance of the rates of resistance to antibiotics indicated for the treatment of an infection and the molecular characterisation of circulating strains may make it possible to identify changes in its epidemiology. In Gipuzkoa (Basque Country, Spain), between 2014 and 2018, the rate of macrolide resistance in *M. genitalium* was <20%, mainly associated with A2058G and A2059G mutations in the 23S rRNA gene and with combined macrolide and fluoroquinolone resistance in <3% of cases [12,13]. In this same period, the distribution of the genetic profiles of *M. genitalium* in Gipuzkoa was diverse [13]. The aims of this study were: (a) to investigate changes in macrolide and fluoroquinolone resistance, as well as the mutations associated with this resistance over the last 3 years (2019–2021), and to compare these data with the previously reported in 2014–2018; (b) to select and molecularly characterise strains considered to be emerging and potentially involved in the trends observed in both periods (2014–2018 and 2019–2021); and (c) to analyse the epidemiological characteristics of these cases to assess whether changes observed might be due to the spread of new clones and/or transmission factors in certain population groups.

## 2. Material and Methods

This study analysed suspected cases of STIs seen in Donostia University Hospital between 2019 and 2021 (San Sebastián, Gipuzkoa) with a catchment population of around 650,000 people, using a commercial molecular technique that detects *C. trachomatis*, *M. genitalium* and *N. gonorrhoeae*, among other microorganisms (Allplex ™ STI Essential Assay, Seegene, Seoul, South Korea). In the cases of infection by *M. genitalium*, the analysis was complemented within 48 h by in-house real-time PCR amplifying the 23S rRNA gene, which detects and differentiates wild macrolide-susceptible strains from strains with resistance-associated mutations [12,14], the results allowing the treatment to be changed as appropriate (to an extended regimen of azithromycin or moxifloxacin, respectively). The mutations were subsequently identified using Sanger sequencing. Fluoroquinolone susceptibility was investigated using conventional amplification and sequencing of *parC* and *gyrA* gene fragments [12,15,16]. In all cases, patients were advised to abstain from unprotected sexual contact until they and their partners had completed treatment, their symptoms had resolved, and their infection test was negative [1]. In cases of treatment failure due to macrolide and fluoroquinolone resistance, other antibiotics were used (doxycycline, pristinamycine, etc.).

When there was remnant DNA for samples found to contain strains considered emerging, the molecular characterisation was completed using the amplification of MG191 (adhesin) and MG309 (lipoprotein) gene fragments, followed by the analysis of the amplified sequences [17,18]. For MG191 fragments, the sequences obtained were compared with known sequences using the Basic Local Alignment Search Tool and aligned using the MEGA 7.0 program to identify the genotype. New genotypes not previously described were deposited in GenBank (accession numbers ON933572-ON933575). The analysis of MG309 fragments was performed by identifying the number of tandem repeats and the localisation of ACT and ATT triplet repeats. The combination of both genotypes allows the strains to be classified as a function of their genetic profiles. Epidemiological data obtained during medical consultations were used to analyse the differences as a function of sex, behaviours and contact networks.

The results of this study (2019–2021) were joined to those of the 2014–2018 period, which were obtained using the same methodology [12,13], in order to assess trends in resistance rates to macrolides and quinolones between the years 2014 and 2021. Over the study, some patients had several different episodes of infection after having received appropriate treatment, and these were considered different if more than 3 months had elapsed between the events. Cases of reinfection and/or recurrence during this period were considered part of the same episode, except in cases in which a new infection was observed after changing a sexual partner.

The statistical analysis was performed using IBM SPSS Statistics (version 23; IBM, Chicago, IL, USA). The database with patient data was anonymised such that individuals could not be identified. The rates of antibiotic resistance and the exact binomial 95% confidence intervals were calculated as functions of the number of episodes in which any mutation associated with macrolide or fluoroquinolone resistance (numerator) was detected and the number of episodes with a valid result on the diagnostic test (denominator). Percentages were compared using chi-square or Fisher exact tests. A *p* < 0.05 was considered statistically significant. To compare variations in the rates of resistance as a function of the type of mutation, the absolute and relative risk increases (ARI and RRI) and the odds ratios (OR) were calculated. The analysis and description of the results of the present study were approved by Donostia University Hospital’s Ethics Committee for Clinical Research (Protocol reference number: DPV-GMG-2018-01 in minutes of September 2018).

## 3. Results

Between 2014 and 2021, a total of 74,809 samples from patients with suspected STIs were analysed, and *M. genitalium* was detected in 1507 (2%, 95% CI 1.9–2.15), corresponding to 1166 episodes of infection. Genetic resistance to macrolides could be studied in 1019 cases (87.4%), to fluoroquinolones in 958 (82.2%) and to both types of antibiotics in 908 (77.9%). The cases in which we did not obtain PCR amplification to identify the type of mutation associated with the corresponding antibiotic resistance mainly corresponded to samples with a low bacterial load (PCR cycle threshold > 35). During the entire period, the overall rates of resistance were 26.1% (95% CI 23.5–28.9%) for macrolides, 10% (95% CI 8.3–12.1) for fluoroquinolones and 6.1% (95% CI 4.7–7.8%) for both antibiotics (Table 1).

The rate of macrolide resistance in *M. genitalium* increased from 17.3% in 2014–2018 to 32.1% in 2019–2021 (*p* < 0.001) (Table 1). This increase between the two periods reached significance both for the A2058G and A2059G mutations (16.6% vs. 27.8%, *p* < 0.001) and the emerging A2058T mutations (0.5% vs. 4.1%, *p* < 0.001), at least one of these mutations being detected in more than 99% of the strains with macrolide resistance (A2059C mutations being found in just two cases). Separately analysing the A2058G and A2059G mutations, a higher increase was observed in the latter. An A2058T mutation was identified in only 2.8% of resistant strains in 2014–2018, the percentage rising to 12.9% in 2019–2021 with a maximum of 28.1% in 2020. The ARI of macrolide resistance was 11.2% for A2058G and A2059G mutations and 3.7% for A2058T mutations, while the RRI was 67.3% and 758.9%, respectively. The OR was 1.9 (95% CI 1.4–2.6) for A2058G and A2059G mutations and 8.9 (95% CI 2.1–37.9) for A2058T mutations. The rate of fluoroquinolone resistance (10%) did not differ significantly between the two study periods but that of resistance to both antibiotics (6.1%) increased significantly (2.9% vs. 8.3%, *p* = 0.001). The more frequently detected mutations were G248T (S83I), and the G259A (D87N) mutations emerged in the second period (0.2% vs. 2.7%. *p* = 0.003). The rate of fluoroquinolone resistance in cases with A2058T mutations (at amino acids 83, 84 and 87 of the QRDR) was 48% vs. 19.5% in cases with other mutations in the 23S rRNA gene (Fisher exact test *p* = 0.001).

Notably, 27 cases of infection by *M. genitalium* resistant to macrolides with A2058T mutations were identified, 2 in 2014–2018 and 25 in 2019–2021. Given this marked increase, the epidemiological and molecular characteristics of these cases were studied. They corresponded to 25 men (24 men who have sex with men [MSM] and 1 heterosexual individual) and 2 women; the majority admitted to having had sexual relationships with various unknown partners, and 9 were HIV positive (Table 2). A total of 19 cases had a history of STIs, and previous treatment with macrolides and/or fluoroquinolones was confirmed in 17. The molecular analysis yielded 8 MG191 (Appendix A) and 4 MG309 (Appendix A) genotypes, the combination describing 15 different genetic profiles (Table 2), and identified 4 new genotypes of the MG191 fragment (genotypes 238–241, Appendix A). The most common genetic profile was number 1, with 8 cases (30%), these being considered a probable network of contacts; further, for another profile (number 2), there were 3 cases (11%) and 1 probable common contact, and for another 3 profiles (numbers 3, 4 and 5), there were 2 cases each (21%, including 2 cases which were probably related [7%]), while the other 10 profiles were each detected in just 1 patient.

## 4. Discussion

Increases in the rates of macrolide resistance in *M. genitalium* infection have been reported in various countries worldwide in recent years [5]. There are several possible reasons for this trend. On the one hand, the single 1 g dose of oral azithromycin, commonly employed as the empirical treatment for non-gonococcal urethritis, results in treatment failures and induces macrolide resistance (20–45%) more often than the targeted treatment with a 500 mg oral dose followed by 250 mg/day for the following 4 days, due to the intracellular concentration of antibiotic being suboptimal for its eradication [19,20]. For this reason, it is key to include *M. genitalium* in the differential diagnosis of suspected STI cases from the outset, to be able to prescribe the appropriate treatment regimen as early as possible. On the other hand, previous treatment with macrolides can also favour the selection of resistant strains as well as the development of resistance by antibiotic selection pressure [21,22]. In line with this, rates of macrolide resistance described in patients seen in STI clinics, with a greater risk of STIs and more previous antibiotic treatments especially MSM, are higher than in the general population seen in primary care centres [2,3,21,23,24]. Additionally, treatment with an adequate regimen of azithromycin in cases of infection of *M. genitalium* susceptible to macrolides can sometimes induce resistance that may lead to recurrence [11,25,26].

In Spain, studies conducted in Barcelona in 2013–2014 and 2016–2017 did not detect a significant increase in the rate of macrolide resistance (35% vs. 36.1%) in patients seen in STI clinics [21,27]. In contrast, in the present study, despite the use of targeted therapy with the extended azithromycin regimen in cases of infection by *M. genitalium* susceptible to macrolides, the overall rate of macrolide resistance significantly increased between 2014–2018 (17%) and 2019–2021 (32%). Further, in the second period, the emergence of A2058T mutations in the 23S rRNA gene of *M. genitalium*, which confer resistance to macrolides (2.8% of resistant cases in 2014–2018 and 12.9% in 2019–2021), was observed. An increase in resistance was found both in cases due to the emerging A2058T mutations (0.5% vs. 4.1%) and in those associated with the more prevalent A2058G and A2059G mutations (16.6% vs. 27.8% for both and 7.0 vs. 13.7 for the latter, respectively). Although the ARI in cases of resistance due to A2058G and A2059G mutations was greater than that due to A2058T mutations (11.2% vs. 3.7%), the RRI due to the emerging A2058T mutations was considerably higher than that of the most prevalent mutations (758.9% vs. 67.3%), and the OR was higher (8.9 vs. 1.9). Therefore, the spread of infection by macrolide-resistant *M. genitalium* with A2058T mutations contributed to increasing the rate of resistance in a context in which an increase was already occurring. Given that the emergence of this mutation coincided with a marked increase in resistance and that it was more often observed among high-risk individuals, we investigated its potential clonal spread.

In all the countries in which these mutations have been studied, the most commonly found are A2058G and A2059G; A2058T and A2059C mutations are uncommon, as was observed in the first part of the study. Notably, A2058T mutations with relative rates of macrolide-resistant strains similarly high to those observed in Gipuzkoa in 2020 (28.1%, 23/133) have only been reported in the Netherlands, with rates of 27.3% (12/44 in 2012–2014) [28] and 21.7% (10/46 in 2014) [29]. Although these findings in the Netherlands could have been attributed to the spread of a clone, in the study by Braam et al., the sequencing of the 23S rRNA gene revealed nucleotide differences between the variant A2058T strains, indicating that this was not a single clonal outbreak [29].

In this study, the molecular characterisation of *M. genitalium* was undertaken through the genetic combination of the MG191 and MG309 alleles. This technique has been shown to have a highly discriminant capacity, having already been used for tracing the infection through networks of sexual contacts and differentiating between persistent and recurrent infections [13,18,30,31]. The strains with an A2058T mutation in the rRNA 23S gene were genetically diverse, and 15 genetic profiles were identified among the 27 cases detected, with only 3 groupings of subjects infected with an apparent relationship, these representing 48% of the cases. Although this diversity is lower than what would be expected in the overall population of cases of *M. genitalium* infection, it probably rules out clonal spread, in line with the results by Braam et al. Further whole-genome sequencing would be useful to facilitate better understanding of the relatedness of these strains within/across sexual networks. Among the strains with an A2058T mutation in the 23S rRNA gene, we observed a higher rate of resistance to fluoroquinolones (48%) than in the strains with other mutations related to macrolide resistance (19.5%), although these strains seemed not be clonal (different mutations at amino acids 83, 84 and 87 of the QRDR). The cases in which this mutation was detected were mainly MSM (24/27) with high-risk sexual behaviours and previous STIs (19/27) treated with macrolides and/or fluoroquinolones, which may have favoured the induction of resistance.

To conclude, the surveillance and monitoring of the molecular epidemiology of *M. genitalium* strains circulating in Gipuzkoa (Spain) enabled us to detect the emergence of A2058T mutations in the 23S rRNA gene associated with macrolide resistance in the second half of the study. Genetic characterisation of the strains harbouring these mutations did not show clonal spread, but clinical and epidemiological data revealed transmission mainly between MSM with high-risk sexual behaviours and a history of STIs. The results indicated that the spread of strains with the emerging A2058T mutations has contributed to the increase in the rate of macrolide resistance observed in the same period, mainly due to an increase in the rates of resistance in cases with the more common A2058G and A2059G mutations.

## Figures and Tables

**Table 1 antibiotics-11-01492-t001:** Comparison of rates of resistance to macrolides (23S rRNA gene) and fluoroquinolones (*parC* and *gyrA* genes) in 1019 episodes of infection by *Mycoplasma genitalium* in Gipuzkoa between 2014 and 2018 (partially described in references [12,13]) and 2019–2021.

Period (Years)	2014–2018 No. (%)	2019–2021 No. (%)	Total No. (%)	*p*
23S rRNA gene PCR positive	415	604	1019	
Resistance-associated 23S rRNA gene mutations	72 (17.3)	194 (32.1)	266 (26.1)	**<0.001**
A2058T	2 (0.5)	25 (4.1)	27 (2.6)	**<0.001**
A2058G	37 (8.9)	66 (10.9)	103 (10.1)	0.29
A2059G	29 (7.0)	83 (13.7)	112 (11.0)	**0.001**
A2058G or A2059G **	3 (0.7)	19 (3.1)	22 (2.2)	**0.008 ***
A2059C	1 (0.2)	1 (0.2)	2 (0.2)	0.651 *
*parC*/*gyrA* gene PCR positive	404	554	958	
Resistance-associated *parC* (+/− *gyrA* ***) gene mutations	35 (8.7)	61 (11.0)	96 (10.0)	0.232
G241T (G81C)	0	1 (0.2)	1 (0.1)	0.874 *
G244A (D82N)	2 (0.5)	0	2 (0.2)	0.347 *
A247C (S83R)	1 (0.2)	1 (0.2)	2 (0.2)	0.623 *
G248A (S83N)	10 (2.5)	9 (1.6)	19 (2.0)	0.351
G248T (S83I) ***	11 (2.7)	19 (3.4)	30 (3.1)	0.535
T250C (F84S)	0	3 (0.5)	3 (0.3)	0.370 *
G259A (D87N)	1 (0.2)	15 (2.7)	16 (1.7)	**0.003** *
G259T (D87Y)	5 (1.2)	3 (0.5)	8 (0.8)	0.418 *
A260G (D87G)	0	1 (0.2)	1 (0.1)	0.874
Others ****	5 (1.2)	9 (1.6)	14 (1.5)	0.622
23S/*parC*/*gyrA* gene PCR positive	376	532	908	
Resistance-associated 23S/*parC*/*gyrA* gene mutations	11 (2.9)	44 (8.3)	55 (6.1)	**0.001**

Statistically significant differences indicated in bold. Chi-square test was used if not otherwise specified. * Fisher exact test. ** In 22 out of 237 (9.3%) cases, resistance-associated 23S rRNA gene mutations were detected using a real-time PCR with probes, but in the subsequent conventional PCR, no amplification was obtained to differentiate between A2058G and A2059G using Sanger sequencing (probably due to a lower bacterial load in this samples). *** 4 cases with G248T (S83I) mutations in *parC* gene and G295 (D99N) mutations in *gyrA* gene. **** G205A (A69T), G213A (G71R), G284A (S95N), C302T (T101I), G355T (A117S), C356A (A117E), C356T (A117V).

**Table 2 antibiotics-11-01492-t002:** Epidemiological and genetic characterisation data for the 27 cases of *Mycoplasma genitalium* infection in carriers of A2058T mutations in the. 23S rRNA gene detected in Gipuzkoa (2014–2021).

Case			Epidemiological Data				Genetic Characterisation			
No.	Type	Date	Sex	Age, Years	Sexual Behaviour, HIV+/not on PrEP, Contacts	Previous STI and Antibiotic Treatments	MG191 Genotype	MG309 Genotype	Allele Pattern	Genetic Profile
6	Vagina	22 October 2019	M	26	MSM, HIV−	No	7	9	7-9	1
8	Pharynx	13 February 2020	M	23	MSM, HIV−, multiple unknown partners	No	7	9	7-9	1
10	Rectum	20 February 2020	M	30	MSM, HIV+, multiple unknown partners	No	7	9	7-9	1
12	Urethra	25 February 2020	M	24	MSM, HIV−, partner of case 11 and other contact with unknown partners	Yes, ceftriaxone, azithromycin	7	9	7-9	1
19	Urethra	10 December 2020	M	34	MSM, HIV+, multiple unknown partners	Yes, doxycycline, moxifloxacin	7	9	7-9	1
20	Rectum	18 December 2020	M	49	MSM, HIV−, PrEP, usual partner negative, but contact with unknown partners	Yes, doxycycline, azithromycin	7	9	7-9	1
24	Urine	8 September 2021	M	30	MSM, HIV−, recent trip to the UK, multiple unknown partners	Yes, azithromycin, moxifloxacin	7	9	7-9	1
27	Pharynx	1 December 2021	M	40	MSM, HIV−, multiple unknown partners	No	7	9	7-9	1
3	Rectum	30 July 2020	M	25	MSM, HIV+, multiple unknown partners; no contact with case 13	Yes, ciprofloxacin, azithromycin, moxifloxacin	238	9	238-9	2
13	Urethra	22 June 2020	M	29	MSM, HIV−, multiple unknown partners; no contact with case 3	Yes, azithromycin, moxifloxacin	238	9	238-9	2
16	Rectum	12 August 2020	M	31	MSM, HIV−; likely contact with cases 3 and 13?	No	238	9	238-9	2
15	Urethra	3 August 2020	M	36	HET, HIV−	No	21	11	21-11	3
17	Urethra	8 October 2020	M	38	MSM, HIV−, usual partner not A2058T carrier, but contact with unknown partners	Yes, azithromycin	21	11	21-11	3
1	Vagina	25 April 2018	W	38	Prostitution	Yes, azithromycin	7	11	7-11	4
18	Urine	30 November 2020	M	47	HET, HIV−	Yes, ceftriaxone	7	11	7-11	4
22	Rectum	27 January 2021	M	29	MSM, HIV+	Yes, azithromycin	21	9	21-9	5
25	Urine	6 August 2021	M	34	MSM, HIV−, PrEP, open relationship	Yes, azithromycin	21	9	21-9	5
2	Cervix	11 April 2018	W	20	HET, HIV−	No	3	10	3-10	6
4	Rectum	4 September 2020	M	29	MSM, HIV+; partners who were not A2058T carriers	Yes, azithromycin, moxifloxacin	239	8	239-8	7
5	Urethra	2 July 2019	M	44	MSM, HIV+	Yes, unknown	240	9	240-9	8
7	Urethra	24 August 2020	M	31	MSM, HIV−, PrEP in Madrid, Madrid contact with unknown partner	Yes, doxycycline, azithromycin, moxifloxacin	241	9	241-9	9
9	Rectum	19 February 2020	M	29	MSM, HIV+	No	2	10	2-10	10
11	Rectum	20 February 2020	M	23	MSM, HIV+, partner of case 12	Yes, azithromycin	7	8	7-8	11
14	Rectum	29 June 2020	M	23	MSM, HIV−, PrEP in Madrid, contact with unknown partner	Yes, azithromycin	7	10	7-10	12
21	Rectum	5 January 2021	M	51	MSM, HIV+, multiple unknown partners	Yes, unknown	241	11	241-11	13
23	Rectum	19 July 2021	M	37	MSM (TSX), HIV−, PrEP	Yes, azithromycin	239	10	239-10	14
26	Pharynx	16 November 2021	M	21	MSM, HIV−, multiple unknown partners	Yes, azithromycin	238	11	238-11	15

M: man; W:woman; HET: heterosexual; MSM: men who have sex with men; TSX: transexual; PrEP: HIV pre-exposure prophylaxis; STI: sexually transmitted infection. Likely related cases are in blue font.

## Data Availability

Not applicable.

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
