# Peer review of "Increases in the Macrolide Resistance of Mycoplasma genitalium and the Emergence of the A2058T Mutation in the 23S rRNA Gene: Clonal Spread?"

_antibiotics, 2022, doi:10.3390/antibiotics11111492_

Round 1
Reviewer 1 Report
Major concern:
I am confusing regarding to the data presentation. As was stated in introduction as well as in the aims - the results of 2019-2021 years will be compared to the results obtained using isolates, isolated between 2014 and 2018; the later have been published (see ref 10). However, the published data (2014-2018) are included in the result’s section as well as in Table 1. The published data should be omitted from the results and all the numbers should be corrected according to the new obtained data; it should be clear to the readers what was done during this specific study. Abstract and other parts of the manuscript should be rewritten accordingly
Minor corrections:
Through the text the name of bacteria/mycoplasma should be italicized; the same is for the genes (parC and gyrA etc.).
L74 – delete “Introduction1”.
The following statements are not clear, please rewrite them/one of them:
L202: is it correct “mutations (758.9% vs 67.3%)”?
Author Response
Answer to Reviewer 1
We would like to thank the reviewer’s comments. Please, find below our response.
Comments and Suggestions for Authors
Major concern:
1) I am confusing regarding to the data presentation. As was stated in introduction as well as in the aims - the results of 2019-2021 years will be compared to the results obtained using isolates, isolated between 2014 and 2018; the later have been published (see ref 10). However, the published data (2014-2018) are included in the result’s section as well as in Table 1. The published data should be omitted from the results and all the numbers should be corrected according to the new obtained data; it should be clear to the readers what was done during this specific study. Abstract and other parts of the manuscript should be rewritten accordingly.
Answer: Previous results in the period 2014-2018 were partially published in reference 10, describing macrolide and fluoroquinolone resistances in 2014-2017, and in reference 11, describing overall macrolide resistance rates and molecular typing of M. genitalium persistent/recurrent infections in 2014-2018 (now these references renumbered as references 12 and 13). Essential components of the surveillance of antibiotic resistance rates are knowing their variation over time and analyzing their molecular basis. In this case, we have analysed the whole data obtained during this first period (2014-2018) and we have compared them with those obtained in the following period 2019-2021. Another aim of the study was to select and molecularly characterise strains considered to be emerging (throughout the two study surveillance periods, from 2014 to 2021), in this case strains with the A2058T mutation, and potentially involved in the differences found, in order to investigate the possible clonal spread of these strains.
In line with the reviewer’s comment and to better clarify this information for the readers, we have made the following changes in the text (listed below):
1.1) Abstract (lines 12-13): “In Gipuzkoa (Basque Country, Spain), the macrolide and quinolone resistance rates in 2014-2018 were have been previously reported as <20% and <10%, respectively”.
1.2) Introduction (line 75): “In Gipuzkoa…, between 2014 and 2018,… in <3% of cases [1012,13].”
1.3) Introduction (lines 78-79): “The aims of this study were… over the last 3 years (2019-2021), and to compare these data with the previously reported in 2014-2018;”
1.4) Material and methods (line 87): “This study analysed suspected cases of STIs seen in Donostia University Hospital between 20142019 and 2021...”
1.5) Material and methods (lines 114-116, new sentence): “The results of this study (2019-2021) were joined to those of the 2014-2018 period, which were obtained using the same methodology [12,13], in order to assess trends in resistance rates to macrolides and quinolones between the years 2014 and 2021.”
1.6) Material and methods (lines 116-117): “Over the 8 years of the study...”
1.7) Table 1 (line 166): “Comparison of rates of resistance to macrolides (23S rRNA gene) and fluoroquinolones (parC and gyrA genes) in 1,019 episodes of infection by Mycoplasma genitalium in Gipuzkoa between 2014-2018 (partially described in references 12 and 13) and 2019-2021.”
Minor corrections:
2) Through the text the name of bacteria/mycoplasma should be italicized; the same is for the genes (parC and gyrA etc.).
Answer: We have corrected this typographic mistake.
3) L74 – delete “Introduction1”.
Answer: We have corrected this formatting error.
4) The following statements are not clear, please rewrite them/one of them:
L202: is it correct “mutations (758.9% vs 67.3%)”?
Answer: Yes, the sentence is correct. The relative risk increase (RRI) results were 758.9% for the emerging A2058T mutations and 67.3% for the most prevalent mutations.
Reviewer 2 Report
This is an interesting study reporting on macrolide resistance mutations in Mycoplasma genitalium. Overall, this is an interesting paper, but does require some additional consideration. Below are some comments/suggestions to improve the publication:
Mycoplasma genitalium must be italicised throughout
Line 40 – the most recent systematic review and meta-analysis on M. genitalium antimicrobial resistance (Machalek et al 2020) indicates that macrolide resistance exceeds 50% across most urban centres globally. It would be useful to revise this and cite this important reference.
Line 46 – 50 – it might also be useful to note here that the use of doxycycline is to lower the M. genitalium load
Line 52 – please also italicise Chlamydia trachomatis
Line 53 – please also italicise Neisseria gonorrhoeae
Line 54 – 55: I think it would be really important here to clarify that global STI management guidelines for M. genitalium indicate only testing of symptomatic patients. It is of course important to include MG in differential diagnoses for symptomatic patients, but I think this needs to be further clarified here.
Lines 58 – 60: It is great to point out that the predictive value for treatment failure with gyrA and parC mutations is variable; but I think it is also important here to reference two other recent papers by Sweeney et al (Lancet Infect Dis 2022) and Vodstrcil et al (Clin Infect Dis 2022) where they point out that there is good predictive value for treatment failure with fluoroquinolones when the parC S83I mutation is present. And conversely, that when parC wildtype sequences are detected, the likelihood of cure with fluoroquinolones is >97%.
Line 74 – 75. I am not sure why you have “.1. Introduction” at the end of this section – perhaps this is a formatting error?
Line 93 – please reword, as “leftover” is not particularly scientific language. Perhaps use the term remnant DNA.
Line 98 – please indicate the genbank accession numbers for new strains deposited within the paper so the readers can identify and access this important strain information.
Line 108 – acknowledging a change in partner is good, but what about if the MG191/MG309 typing results were different (ie indicating a different infecting strain?). Was this also investigated?
For Table 1, can you please stratify results per each resistance mutation. You have already done this for A2058T and A2059C, but it would be really useful to see this data separately for A2058G and A2059G too.
Within your methods section, you note that parC and gyrA sequencing was performed, but you do not actually present any substantial results for this in the paper which is really disappointing. Can you please present these results formally within the results section, and also specify the parC or gyrA mutations found in the QRDR of these genes in Table 2, for transparency and clarity. Since it is mentioned in your methods section, I was expecting to see this data.
Table 2. There is a red line through cell 19, please fix this formatting issue. This looks to be a picture copied from an excel spreadsheet, can you please input this into a proper table for the final paper please.
You could also consider presenting this data in table 2 as a phylogenetic tree to demonstrate the relatedness of strains based on the MG191/MG309 amplicons, then overlay other information such as sexual behaviour/antibiotic use on top of that to demonstrate the relatedness of the presumed sexual networks here too.
Line 175 – 176: please note the prevalence of treatment failures due to induced resistance.
Please reword the sentence on lines 208 - 209 to make this clearer
Comment on sentence on lines 213 – 215 and 224 - 226: This finding is only as good as the typing scheme in place for MG and I think you are placing a lot of weight on the fact we are sequencing two highly variable regions of the genome and stating that these are very different strains. I think perhaps you need to temper the phrasing of this. Although Braam et al supports your finding, I think the limitation here is we are targeting only two very variable regions of the genome to make this claim, so further WGS would be useful here to determine the overall relatedness of these strains too. Please perhaps indicate the utility of WGS to facilitate better understanding of the relatedness of these strains within/across sexual networks.
Line 227 – you state there was a higher rate of fluoroquinolone resistance but you do not look at specific mutations in parC and gyrA here, like you have done for macrolides. Please present this data.
Author Response
Answer to Reviewer 2
We would like to thank the reviewer’s comments. Please, find below our response.
Comments and Suggestions for Authors
This is an interesting study reporting on macrolide resistance mutations in Mycoplasma genitalium. Overall, this is an interesting paper, but does require some additional consideration. Below are some comments/suggestions to improve the publication:
1) Mycoplasma genitalium must be italicised throughout.
Answer: We have corrected this typographic mistake.
2) Line 40 – the most recent systematic review and meta-analysis on M. genitalium antimicrobial resistance (Machalek et al 2020) indicates that macrolide resistance exceeds 50% across most urban centres globally. It would be useful to revise this and cite this important reference.
Answer: This excellent review was cited in the sentence alluded (reference 5). However, we would like to maintain the figure cited in the text (>40%), since in the WHO European Region, the prevalence of macrolide resistance tends to be somewhat lower than the summary prevalence found globally. For example, the prevalence of resistance-associated mutations for azithromycin reported by Nijhuis et al in the Netherlands (2018-2019) was 40.6% (reference 6).
3) Line 46 – 50 – it might also be useful to note here that the use of doxycycline is to lower the M. genitalium load.
Answer: We agree with a more detailed explanation of doxycycline use. We have added a new sentence in lines 60-64: “Although doxycycline therapy has limited cure rates (~30%), sequencial targeted therapy could be a useful strategy in settings with high macrolide resistance rates and if a macrolide susceptibility test could not be performed early, allowing the initial use of doxycycline to lower the M. genitalium load [5].”
4) Line 52 – please also italicise Chlamydia trachomatis.
Answer: We have corrected this typographic mistake.
5) Line 53 – please also italicise Neisseria gonorrhoeae.
Answer: We have corrected this typographic mistake.
6) Line 54 – 55: I think it would be really important here to clarify that global STI management guidelines for M. genitalium indicate only testing of symptomatic patients. It is of course important to include MG in differential diagnoses for symptomatic patients, but I think this needs to be further clarified here.
Answer: Following this suggestion and to clarify this point, we have added in this sentence (line 57) the key word “symptomatic” before “STIs”.
7) Lines 58 – 60: It is great to point out that the predictive value for treatment failure with gyrA and parC mutations is variable; but I think it is also important here to reference two other recent papers by Sweeney et al (Lancet Infect Dis 2022) and Vodstrcil et al (Clin Infect Dis 2022) where they point out that there is good predictive value for treatment failure with fluoroquinolones when the parC S83I mutation is present. And conversely, that when parC wildtype sequences are detected, the likelihood of cure with fluoroquinolones is >97%.
Answer: We agree. We have included these two references (new references 10 and 11, respectively), and pointed out the existence of different predictive values depending on the mutations by inserting a new sentence (lines 67-69): “G248T mutations in the parC gene (S83I aminoacid change) have been associated with higher rates of moxifloxacin treatment failures, while the absence of mutations at parC S83/D87 was highly predictive of cure (>98%).”
We have changed in the text the numbering of the remaining references accordingly.
8) Line 74 – 75. I am not sure why you have “.1. Introduction” at the end of this section – perhaps this is a formatting error?
Answer: We have corrected this formatting error.
9) Line 93 – please reword, as “leftover” is not particularly scientific language. Perhaps use the term remnant DNA.
Answer: We have replaced “leftover” with “remnant” (line 102).
10) Line 98 – please indicate the genbank accession numbers for new strains deposited within the paper so the readers can identify and access this important strain information.
Answer: Following this recommendation, we have moved this information from the “Results” section (line 187) to the “Material and methods” section (line 108). Genbank accession numbers are also included in the Supplementary Table S1.
11) Line 108 – acknowledging a change in partner is good, but what about if the MG191/MG309 typing results were different (ie indicating a different infecting strain?). Was this also investigated?
Answer: Molecular typing of M. genitalium strains was not performed in all infection cases. This laborious work could only be carried out in the context of research, i.e. for discriminating between persistent and recurrent infections in cases of treatment failure (reference 13), or in the present study to rule out clonal spread of M. genitalium strains carrying the A2058T mutation.
12) For Table 1, can you please stratify results per each resistance mutation. You have already done this for A2058T and A2059C, but it would be really useful to see this data separately for A2058G and A2059G too.
Answer: In line with this reviewer’s suggestion, we have incorporated in table 1 a separately analysis of the A2058G and A2059G mutations. We have also noted that in 22 out of 237 cases the specific identification of the mutation was not reached using the Sanger sequencing (this point has been specified in the footnotes of the table). Accordingly, in the text we have added the following sentences:
Results (lines 150-151): “Analysing separately the A2058G and A2059G mutations a higher increase was observed in the latter.”
Discussion (line 226): “…for both and 7.0 vs 13.7 for the latter, respectively”.
13) Within your methods section, you note that parC and gyrA sequencing was performed, but you do not actually present any substantial results for this in the paper which is really disappointing. Can you please present these results formally within the results section, and also specify the parC or gyrA mutations found in the QRDR of these genes in Table 2, for transparency and clarity. Since it is mentioned in your methods section, I was expecting to see this data.
Answer: We suppose the reviewer is referring to table 1. We have now completed in table 1 the results about parC and gyrA pointing out the specific mutations detected. In the text we have added the following sentences:
Results (lines 158-160): The more frequently detected mutations were G248T (S83I), and the G259A (D87N) mutations emerged in the second period (0.2% vs 2.7%. p=0.003).
Discussion (line 257-259): “…, although these strains seemed not be clonal (different mutations at amino acids 83, 84 and 87 of the QRDR)”.
14) Table 2. There is a red line through cell 19, please fix this formatting issue. This looks to be a picture copied from an excel spreadsheet, can you please input this into a proper table for the final paper please.
Answer: We have solved the problem with the red line in the edition of table 2.
15) You could also consider presenting this data in table 2 as a phylogenetic tree to demonstrate the relatedness of strains based on the MG191/MG309 amplicons, then overlay other information such as sexual behaviour/antibiotic use on top of that to demonstrate the relatedness of the presumed sexual networks here too.
Answer: Following the suggestion of another reviewer we have modified table 2, now being the information about the relatedness of strains clearer for the reader. We think that a phylogenetic tree would not improve this view and it would have less epidemiological information than the new table2.
16) Line 175 – 176: please note the prevalence of treatment failures due to induced resistance.
Answer: Following this suggestion, in line 203, we have noted the prevalence of treatment failures due to the use of 1 g azithromycin described in references 19 and 20: (20-45%).
17) Please reword the sentence on lines 208 - 209 to make this clearer.
Answer: We have slightly modified this sentence to make it clearer for the reader. Now it appears in lines 235-237: “In all the countries in which these mutations have been studied, the most commonly found are A2058G and A2059G; A2058T and A2059C mutations are uncommon, as was observed in the first part of the study.”
18) Comment on sentence on lines 213 – 215 and 224 - 226: This finding is only as good as the typing scheme in place for MG and I think you are placing a lot of weight on the fact we are sequencing two highly variable regions of the genome and stating that these are very different strains. I think perhaps you need to temper the phrasing of this. Although Braam et al supports your finding, I think the limitation here is we are targeting only two very variable regions of the genome to make this claim, so further WGS would be useful here to determine the overall relatedness of these strains too. Please perhaps indicate the utility of WGS to facilitate better understanding of the relatedness of these strains within/across sexual networks.
Answer: In a previous study in Gipuzkoa, using the same typing method (MG191/MG309 genes), we described a very broad distribution of genetic profiles in M. genitalium infections (reference 13). The typing method performed in this study is much more discriminative than that used by Braam et al. (23S rRNA gene). So, if we had detected the same genetic profile in the strains, we would confirmed it was a clonal spread, that was our main hypothesis. Conversely, we can’t assure that a different genetic profile correspond to a “very" different strain, but we don’t affirm this in the text.
Nevertheless, following the reviewer’s suggestion, we have tempered the phrasing in the text adding the word “probably” (line 252), and we have indicated the possible utility of WGS in a new sentence (lines 253-255): “Further whole genome sequencing would be useful to facilitate better understanding of the relatedness of these strains within/across sexual networks.”
19) Line 227 – you state there was a higher rate of fluoroquinolone resistance but you do not look at specific mutations in parC and gyrA here, like you have done for macrolides. Please present this data.
Answer: Please see comment number 13.
Reviewer 3 Report
This well written article describes the monitoring of genetically determined antibiotic resistance to macrolides and fluoroquinolones in Mycoplasma genitalium (MG). It contains data on large sample sets of two calendar periods (2014-2018 and 2019-2021) in Basque country. All macrolide associated mutations in the 23SrRNA region increased but the fluoroquinolone resistance mutations in gyrA/parC genes did not. The single nucleotide polymorphism (SNP) A2058T mutation prevalence in the 23SrRNA gene also significantly increased. Next to determination of antibiotic resistance SNPs also parts of two polymorphic genes, MG191adhesin and MG309 lipoprotein, were sequenced to look at possible clonal expansion of the MG strains.
All these data are very interesting.
There are a few points that need to be answered to further improve the quality of this paper and make it suitable for publication.
Major and minor points.
- The title of the manuscript does not seem to fit with the actual message, especially the part ‘causality or coincidence’. There are no experiments that support either causality or coincidence. Maybe the authors can use the words: ‘possible clonality’? Or ‘clonal spread’?
- Throughout the manuscript the bacterial names should be written in italics: Mycoplasm genitalium, Chlamydia trachomatis, Neisseria gonorrhoeae. Also gene names, such as gyrA and parC need to be written in italics. Please adjust.
- Introduction: the sentence ‘In recent years……treated with azithromycin [2-6]’ is very long and therefor not easy to understand. Please split up this sentence to make the message more comprehensive.
- Methods, line 74/75: ‘1. Introduction’ should be removed there.
- Results, lines 124-126: ‘susceptibility to macrolides….’: it would be more informative to provide the ‘genetic resistance’ since this is the focus of this study. Please adjust.
- Table 2 is printed very small. Please try to enlarge. Also: why are some rows in blue? This is not mentioned in the table legend.
In addition, in Table 2, the order of presenting the 27 cases with A2058T mutation seems random. Is it on the genetic profile? But these profiles were self-chosen. Please reorganise this table; for example present the data in order of the largest cluster first and single types later and then ordered according to patient number.
Author Response
Answer to Reviewer 3
We would like to thank the reviewer’s comments. Please, find below our response.
Comments and Suggestions for Authors
This well written article describes the monitoring of genetically determined antibiotic resistance to macrolides and fluoroquinolones in Mycoplasma genitalium (MG). It contains data on large sample sets of two calendar periods (2014-2018 and 2019-2021) in Basque country. All macrolide associated mutations in the 23SrRNA region increased but the fluoroquinolone resistance mutations in gyrA/parC genes did not. The single nucleotide polymorphism (SNP) A2058T mutation prevalence in the 23SrRNA gene also significantly increased. Next to determination of antibiotic resistance SNPs also parts of two polymorphic genes, MG191adhesin and MG309 lipoprotein, were sequenced to look at possible clonal expansion of the MG strains.
All these data are very interesting.
There are a few points that need to be answered to further improve the quality of this paper and make it suitable for publication.
Major and minor points.
1) The title of the manuscript does not seem to fit with the actual message, especially the part ‘causality or coincidence’. There are no experiments that support either causality or coincidence. Maybe the authors can use the words: ‘possible clonality’? Or ‘clonal spread’?
Answer: Following the reviewer suggestion we have changed the title, and replaced ‘causality or coincidence?’ with ‘clonal spread?’. The new title now is “Increases in macrolide resistance of Mycoplasma genitalium and emergence of the A2058T mutation in the 23S rRNA gene: clonal spread?”
2) Throughout the manuscript the bacterial names should be written in italics: Mycoplasm genitalium, Chlamydia trachomatis, Neisseria gonorrhoeae. Also gene names, such as gyrA and parC need to be written in italics. Please adjust.
Answer: We have adjusted these typographic mistakes.
3) Introduction: the sentence ‘In recent years……treated with azithromycin [2-6]’ is very long and therefor not easy to understand. Please split up this sentence to make the message more comprehensive.
Answer: Following this suggestion and for a better understanding, we have splitted up the sentence and slightly modified it (line 42).
4) Methods, line 74/75: ‘1. Introduction’ should be removed there.
Answer: We have corrected this typographic mistake.
5) Results, lines 124-126: ‘susceptibility to macrolides….’: it would be more informative to provide the ‘genetic resistance’ since this is the focus of this study. Please adjust.
Answer: We have adjusted the sentence replacing ‘susceptibility...was confirmed’ with ‘genetic resistance...could be studied’ (lines 137-138).
6) Table 2 is printed very small. Please try to enlarge. Also: why are some rows in blue? This is not mentioned in the table legend. In addition, in Table 2, the order of presenting the 27 cases with A2058T mutation seems random. Is it on the genetic profile? But these profiles were self-chosen. Please reorganise this table; for example present the data in order of the largest cluster first and single types later and then ordered according to patient number.
Answer: According to these instructions, we have adapted table 2, and modified in the text the new genetic profile numbers (lines 188-191).
Round 2
Reviewer 1 Report
I do not have.